# Theoretical Studies of Nickel-Dependent Enzymes

**Per E. M. Siegbahn [1],\*, Shi-Lu Chen [2]  and Rong-Zhen Liao [3]**

[1] Department of Organic Chemistry, Arrhenius Laboratory, Stockholm University, SE-10691 Stockholm, Sweden

[2] School of Chemistry and Chemical Engineering, Beijing Institute of Technology, Beijing 100081, China

[3] School of Chemistry and Chemical Engineering, Huazhong University of Science and Technology, Wuhan 430074, China

\* Correspondence: per.siegbahn@su.se

**Abstract:** The advancements of quantum chemical methods and computer power allow detailed mechanistic investigations of metalloenzymes. In particular, both quantum chemical cluster and combined QM/MM approaches have been used, which have been proven to successfully complement experimental studies. This review starts with a brief introduction of nickel-dependent enzymes and then summarizes theoretical studies on the reaction mechanisms of these enzymes, including NiFe hydrogenase, methyl-coenzyme M reductase, nickel CO dehydrogenase, acetyl CoA synthase, acireductone dioxygenase, quercetin 2,4-dioxygenase, urease, lactate racemase, and superoxide dismutase.

**Keywords:** nickel enzymes; reaction mechanism; quantum chemical calculations

## 1. Introduction

Nature harnesses transition metals to catalyze many different types of biological reactions that are crucial to life. Although about one-third of the enzymes are metalloenzymes, the use of nickel as a cofactor is not very common [1–9]. It was in 1975 that Zerner discovered the first nickel-dependent enzyme, namely urease [10]. Until now, ten different types of nickel enzymes (Table 1) have been reported with diverse biological functions. Among the known nickel enzymes, four of them are involved in the processing of anaerobic gases, namely, $H_2$, CO, $CO_2$, and $CH_4$, which have been proposed to be crucial to the origin of life [7]. Ni–Fe hydrogenase mediates the reversible two-electron reduction of protons to $H_2$ [11]. Methyl-coenzyme M reductase catalyzes the reversible transformation of methyl-coenzyme M plus coenzyme B into $CH_4$ and heterodisulfide CoM–S–S–CoB [12]. CO dehydrogenase catalyzes the reversible oxidation of CO to $CO_2$ using water as the oxygen source [13]. Acetyl-CoA synthase interacts tightly with CO dehydrogenase and uses the CO generated by CO dehydrogenase, CoA-SH, and a methyl group from a corrinoid/FeS protein to synthesize acetyl-CoA [14]. Acireductone dioxygenase [15], and quercetin 2,4-dioxygenase [16] are both nickel-dependent dioxygenases, and produce CO as one of the products. Ni-dependent superoxide dismutase targets the toxic reactive oxygen species superoxide and catalyzes its conversion to $O_2$ and $H_2O_2$ [17]. All these enzymes belong to the oxidoreductase class, while the remaining three enzymes are involved in the hydrolysis of urea (urease) [10], the isomerization of methylglyoxal to lactate (glyoxylase I) [18], and the racemization of lactate (lactate racemase, or LarA) [19].

Understanding the reaction mechanisms of nickel-dependent enzymes is of fundamental and practical importance. A particularly interesting question is why these enzymes select nickel as a catalytic center. One important advantage of using nickel has been suggested to be related to its flexible coordination geometry [8]. For redox reactions, the ligand environment has been demonstrated to play a crucial role in tuning the redox potentials of nickel, ranging from +0.89 V in superoxide

dismutase to −0.60 V in Methyl-coenzyme M reductase and CO dehydrogenase [6]. Sulfur-donor ligands, such as sulfides, thiolates of cysteine residues, are commonly used to fulfill this purpose. In addition, the Lewis acidic character of nickel has also been suggested to be important for urease, glyoxalase I, acireductone dioxygenase, and quercetin 2,4-dioxygenase [8]. Importantly, these enzymes preferentially adopt O/N-donor ligands. Another important question is how the enzyme catalyzes the reaction. To understand the mechanistic details, especially the structures of transition states and intermediates for all elementary steps, quantum chemical calculations have proven to be useful to complement experimental work.

**Table 1.** Reactions catalyzed by Ni-dependent enzymes.

Two popular but different approaches have been developed in the modeling of metalloenzymes. The first one is termed the quantum chemical cluster approach, developed by Siegbahn and Himo et al. [20–26]. More than 30 years of experience has demonstrated that a relatively small model of the active site, from about 30 atoms initially to more than 200 atoms nowadays, is capable of describing all important mechanistic features of the catalysis. The alternative approach is called the combined quantum mechanics/molecular mechanics (QM/MM) method, which was first proposed by Warshel and Levitt in 1976 [27]. It has been shown by Liao and Thiel that with proper selection and an increase of the number of atoms in the QM region, both approaches gave similar results and the same conclusion [28,29].

In the present review, we summarize the mechanistic studies on nickel enzymes by quantum chemical calculations.

## 2. NiFe Hydrogenase

In nature, hydrogenases are the enzymes designed to form dihydrogen from protons and electrons in a reversible process:

$$2H^+ + 2e^- \rightleftharpoons H_2 \tag{1}$$

The most common forms of hydrogenases are NiFe- and FeFe-hydrogenases. The NiFe-enzymes are primarily used for hydrogen oxidation, and the FeFe-enzymes for proton reduction. NiFe-hydrogenase is probably the Ni-enzyme that has been most intensively studied by theoretical methods. The active site [30] is shown in Figure 1. Apart from the four cysteines, there are three additional ligands on iron, two cyanides, and one carbon monoxide, which are very unusual in biological systems, since they are potentially poisonous. A complicated machinery involving many enzymes is, therefore, used to construct this complex.

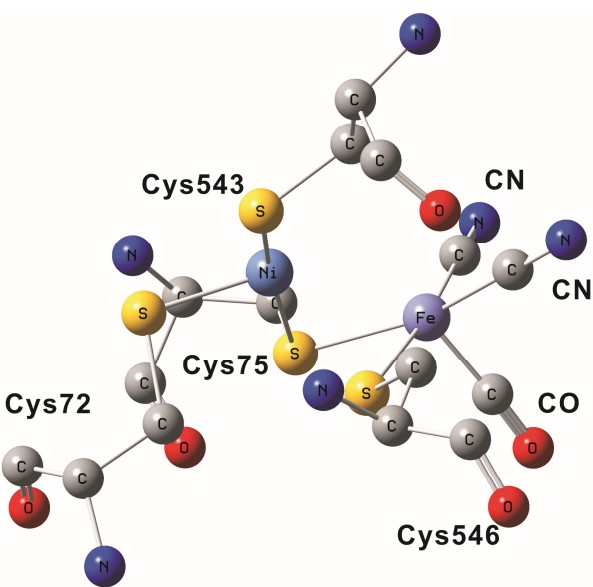

**Figure 1.** The X-ray structure of the active site of NiFe hydrogenase (Protein Data Bank (PDB) code: 1YRQ) [30].

The theoretical work prior to 2007 has been reviewed [31,32]. A variety of different models had been used, ranging from a minimal one, with only the direct ligands of the NiFe complex, to larger ones, with some charged second shell residues. All studies agreed on the general mechanism described in Figure 2. It contains three intermediate states observed experimentally, $Ni_a$–C*, $Ni_a$–S and $Ni_a$–SR. $Ni_a$–C* is the electron paramagnetic resonance (EPR)-active resting state, which has a bridging hydride between the metals. The EPR-silent $Ni_a$–SR state is reached by reduction of $Ni_a$–C*, while $Ni_a$–S, which is also EPR-silent, is obtained by oxidation. To complete the cycle, $Ni_a$–SR can be reached by adding dihydrogen to $Ni_a$–S. Several other states have also been observed experimentally under varying conditions. The TS for H–H cleavage was generally described as heterolytic, leading to a bridging hydride and a protonation of a cysteine, usually Cys543. This mechanism could be regarded as being in consensus among the theoretical groups [31,32], and had strong support from experiments. For the energy profile, it could be concluded that entropy and dispersion played important roles [31], even though the dispersion effect could not be routinely included in the calculations, as it can be now.

**Figure 2.** Schematic drawing of the catalytic cycles for NiFe-hydrogenases, suggested by experiments. Adapted from [31], Copyright © 2007, American Chemical Society.

In the years following the early period before 2007, some new possibilities and aspects were investigated. This development has also been briefly reviewed [26,33]. First, from experiments, it was suggested that a variant of the mechanism in Figure 2 might be operative, leading to a mechanism in Figure 3 [34]. The main difference to the mechanism of Figure 2 described above was that $Ni_a$–S does not participate in the active cycle. Instead, catalysis starts only after an initial reaction between $Ni_a$–S and dihydrogen, where $Ni_a$–C* is created. $Ni_a$–C* is then only in equilibrium with $Ni_a$–SR during turnover. Some support for the alternative mechanism was given in one study [35]. An interesting state in the context of that mechanism was a Ni(I)-state ($Ni_a$–R*), appearing at the end of the single step heterolytic mechanism. A TS for oxidative addition was found leading to two hydrides, one bridging and one bound to nickel ($Ni_a$–X*).

Another development was that the interest switched over to the oxidized states. The reason for this interest is the possibility to generate fuel by combining water oxidation with $H_2$ formation. A major problem in this context is that most hydrogenases are inactivated with $O_2$ present. Two main oxidized states of the NiFe cofactor have been identified. In Ni–B, there is a bridging hydroxide between Ni(III) and Fe(II). It is inactive for $H_2$ cleavage, but is relatively easy to reactivate. The Ni–A state is more difficult to reactivate. At the highest resolution (1.1 Å) it was initially concluded that Ni–A is a peroxide with one oxygen bridging between the metals and the other oxygen on nickel [30,36]. The early modeling studies were reviewed in 2007 [31]. While there was no problem obtaining the Ni–B structure with a bridging hydroxide [37,38], the modeling of Ni–A was more problematic. A structure similar to the one found experimentally could rather easily be obtained, but the energy was too high. A detailed analysis of the density of the experimental structure showed that it was likely to be a mixture of states, and the structure was probably also over reduced [39]. Calculations on the mechanism for the reactivation of the initially suggested structure for Ni–A were made [37]. It was suggested that the full reactivation of the enzyme requires two additional electrons and protons and this is the reason that it was slow. A reactivation mechanism for an oxidized cysteine ligand was also investigated. In 2015, a

re-analysis of the experimental structure for Ni–A was performed, and it was concluded that Ni–A instead has a bridging hydroxide and an oxidized sulfenated form of Cys75 [40]. DFT calculations were then done for the new structure using larger models than before [41,42], and a revised mechanism for the reactivation was made.

**Figure 3.** A variant of the mechanism for NiFe-hydrogenases. Adapted from [31], Copyright © 2007, American Chemical Society.

Several oxygen insensitive NiFe hydrogenases have been found. In these enzymes, the Ni–A state is activated by two electrons supplied from the proximal FeS cluster [43]. A few DFT studies have been performed to determine the details of the mechanism of how the electrons are released from the FeS cluster [44–46]. The oxygen tolerant hydrogenases have been found to have specifically designed proximal Fe–S clusters, where a sulfide in a corner is missing and replaced by two cysteines. This allows the Fe–S cluster to deliver two electrons to the NiFe-cluster. Together with two electrons from the active site, the accumulation of the Ni–A state is diminished. DFT calculations were used to determine electronic and structural changes of the Fe–S cluster.

After these initial modeling studies, the interest in the mechanism of NiFe-hydrogenase has decreased. The studies mentioned above gave mechanisms that are quite similar with minor differences, and in good agreement with available experiments [31]. However, in recent years there was a set of studies by Ryde et al. in 2014–2018, where many of the results obtained previously were seriously criticized [47–51]. In particular, the accuracy of the Becke, 3-parameter, Lee–Yang–Parr exchange-correlation functional (B3LYP), used in practically all previous studies, was claimed to be insufficient. Comparisons between B3LYP and more advanced methods of ab initio type seemed to give results, showing that there were large errors using B3LYP. For the bare cofactor without hydrogen, the singlet–triplet splitting (S–T) error was tolerable with a usual error for B3LYP of a few kcal/mol compared to the advanced methods [47]. However, when $H_2$ was added, the S–T splitting showed large errors for B3LYP [48], with a value of 13.2 kcal/mol compared to high level studies that gave 2–3 kcal/mol [49], but there are severe problems with that comparison. The most important one is that the structures were not optimized, but taken directly from a QM/MM study. This meant that the structures, as used in the comparison, are about 40 kcal/mol higher in energy than the optimized ones. The result most emphasized in the new DFT studies was that $Ni_a$–R* should not be an intermediate in the reaction sequence [51], as it was in some previous studies [35]. However, there is an interesting experimental result in this context using electro-chemical methods, where the conclusion was that *$Ni_a$–R** was indeed observed as an intermediate [52]. Another more important aspect of the Ryde et al. studies is that they showed that it is actually possible to do calculations with the most accurate methods in quantum chemistry, such as the coupled cluster single-double and perturbative triple method (CCSD(T)) and the density matrix renormalization group method (DMRG). They have not yet been used for the reaction mechanism but it will be interesting to follow that development in the future.

## 3. Methyl-Coenzyme M Reductase

Methane, acting as the second most important anthropogenic greenhouse gas after $CO_2$ [53,54], has a significant influence on the global carbon cycling and the climate [53]. With a methane emission annually of around 500–600 Tg, about 69% is biologically released by microbial metabolisms, i.e., methanogenic archaea [55]. Meanwhile, a large amount of methane is oxidatively consumed by aerobic methanotrophic bacteria and anaerobic methane oxidizing archaea (ANME) [56,57]. The anaerobic methane oxidation is usually coupled with the reduction of sulfate (sulfate-reducing δ-proteobacteria) [58,59] iron [60,61], manganese [61], nitrate [62], soluble metal complexes [63], etc. [64], which may critically affect the Proterozoic and Neoproterozoic global climate systems [65] and the breeding of animals [66]. The final step of microbial methane formation and the first step of anaerobic methane oxidation are both performed by the Ni-dependent methyl-coenzyme M reductase (MCR) [56,67], which is capable of reversibly catalyzing the conversion of methyl-coenzyme M ($CH_3$–SCoM) and coenzyme B (CoB–SH) to a heterodisulfide (CoM–S–S–CoB) and methane using a Ni(I) porphyrinoid cofactor ($F_{430}$) (Figure 4) [6,9]. It is further found that MCR-like enzymes may constitute a family of alkyl-coenzyme M reductases, which are essential for the microbial anaerobic oxidation of ethane [68], propane [56,68], and butane [69] following the same reversible reaction pattern as MCR, i.e., alkane + CoM–S–S–CoB ⇌ alkyl–SCoM + CoB–SH. Thus, the MCR mechanism may serve as a template for the family of alkyl-coenzyme M reductases [68], rendering it significant for the understanding of the alkane biosynthesis and the development of biomimetic catalysts for the alkane conversion.

**Figure 4.** The reaction catalyzed by methyl-coenzyme M reductase (MCR) using a $F_{430}$ cofactor.

Mainly four mechanisms for MCR have been proposed and examined by experiments or theoretical calculations (Figure 5). Accompanying the crystallization of MCR [12], mechanism I (Figure 5a) was hypothesized to include an organometallic Ni(III)-methyl intermediate (referred to as $MCR_{Me}$) resulting from the Ni-activated C–S bond dissociation of $CH_3$–SCoM and the proton transfer from CoB–SH to the sulfur of $CH_3$–SCoM. In mechanism I, the first step of C–S bond dissociation is most likely not rate-limiting, since the following electron transfer was suggested to lead to an unstable CoM–S·H radical. Methane is formed in the next step of S–S bond formation. This mechanism received the strongest support from the direct synthesis of $MCR_{Me}$ from the active MCR ($MCR_{red1}$) with methyl bromide [70] or iodide [71], although its formation from the native substrate ($CH_3$–SCoM) has never been found. However, a rough estimation by means of the B3LYP bond dissociation energies (BDEs) predicted a very large endothermicity for the $MCR_{Me}$ formation in the native MCR reaction [72]. A more elaborate B3LYP investigation based on a chemical model built from the crystal structure (Figure 6) showed that the $MCR_{Me}$ formation is dependent on the acidity of the methyl-X leaving group, i.e., the C–X bond dissociation energy [73]. The $MCR_{Me}$ formation is facile with a barrier of a few kcal/mol when X is a halide ($I^-$, $Br^-$, or $Cl^-$), while it has a very high barrier when starting from the native substrate with a stronger C–S bond, which was predicted to have a large endothermicity of 23.5 kcal/mol [73]. This endothermicity was then updated to 21.8 kcal/mol with dispersion and entropy

effects added [74], which is sufficient to rule out mechanism I, considering the possible additional barriers in the subsequent steps (proton transfer and electron transfer).

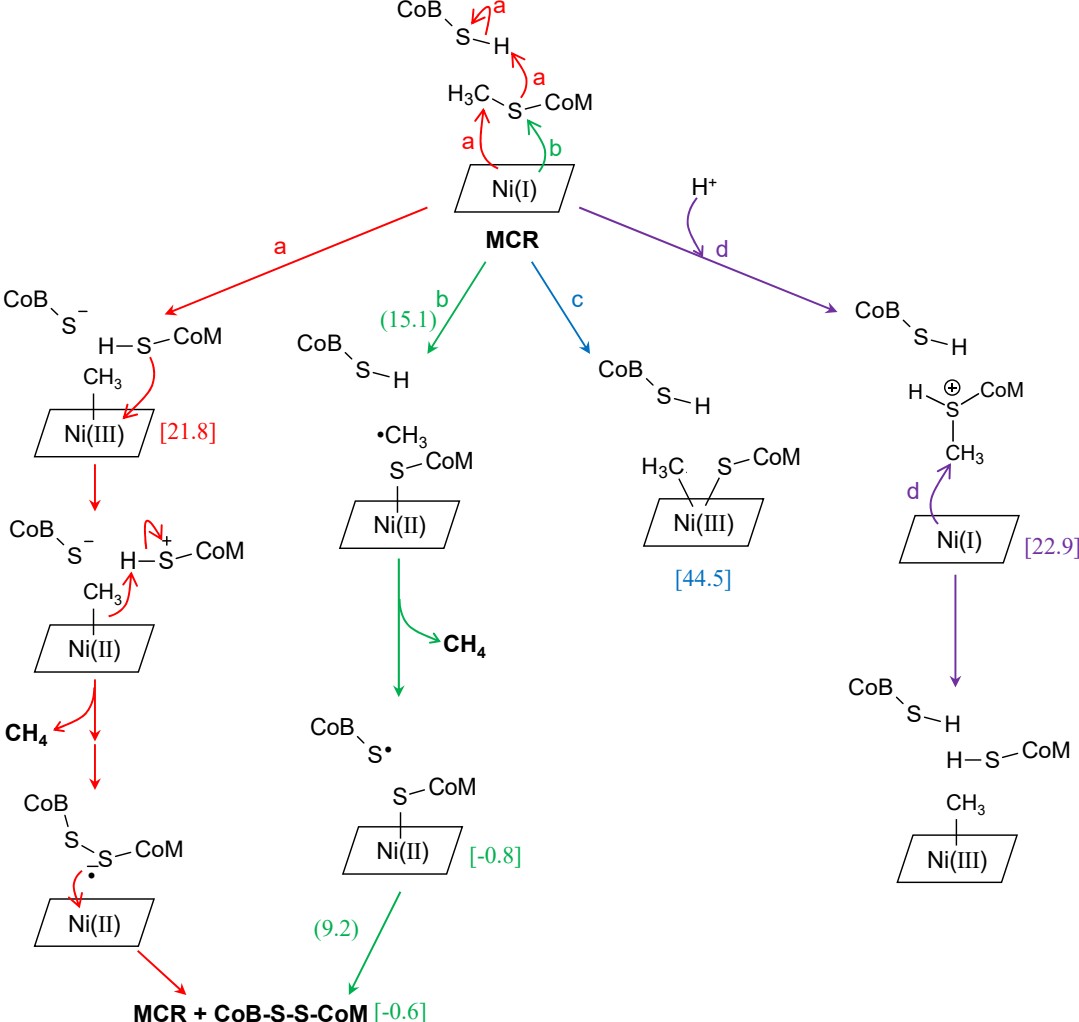

**Figure 5.** Four proposed mechanisms for the MCR reaction. (**a**) Mechanism I, including a methyl-Ni(III)F$_{430}$ intermediate. (**b**) Mechanism II, involving a methyl radical and a CoM–S–Ni(II)F$_{430}$ intermediate. (**c**) Mechanism III ,including a ternary intermediate with side-on methyl and CoM–S$^-$ coordinating to Ni. (**d**) Mechanism IV, involving a proton binding at the sulfur of CH$_3$–SCoM. The energies for minima and transition states are given in brackets and parentheses, respectively. The energies for mechanisms I and II are adapted from [74], copyright © WILEY-VCH Verlag GmbH & Co. KGaA, Weinheim, while the ones for mechanisms III and IV are adapted from [75], with permission from the PCCP Owner Societies.

A more detailed DFT cluster modeling of mechanism II (Figure 5b) was then performed [72,76]. Mechanism II proceeds through a C–S bond dissociation forming Ni(II)–SCoM and a methyl radical, the immediate quenching of the methyl radical by CoB–SH leading to methane and a CoB–S· radical. This is followed by a rebound mechanism between Ni(II)–SCoM and CoB–S· to form a CoM–S–S–CoB and a regeneration of Ni(I) F$_{430}$. This mechanism has the following characteristics that are compatible with most of the mechanistic experiments: (i) The first step of C–S bond dissociation is rate-limiting with a barrier of 19.5 kcal/mol, which is supported by a substantial carbon kinetic isotope effects ($^{12}$CH$_3$–SCoM/$^{13}$CH$_3$–SCoM) of 1.04 ± 0.01 [77]. (ii) The methyl radical is captured rapidly by CoB–SH, so that methane formation happens almost simultaneously with the C–S bond dissociation and the whole MCR reaction can be regarded as two chemical steps via two transition states. (iii) An inversion

of configuration at the methyl carbon takes place via a trigonal planar configuration during the methane formation; step (iii) combined with step (ii) is consistent with the finding using an isotopically chiral form of ethyl-coenzyme M [78] and a large $\alpha$-secondary kinetic H/D isotope effect ($k_\text{H}/k_\text{D} = 1.19 \pm 0.01$), which indicates a geometric change of the methyl group from tetrahedral to trigonal planar at the highest transition state [77]. However, the energetics obtained in 2002 and 2003 [72,76] are not accurate enough to give a quantitatively correct description of the MCR mechanism, mainly because of the use of methanol as the model of Tyr333 and Tyr367, the lack of dispersion effects, and the omission of entropy effects in the second step of the S–S bond formation [74]. This leads to an excessively high barrier of ~26 kcal/mol for the reverse reaction (i.e., the anaerobic oxidation of methane) and an excessively large barrier difference of 10.0 kcal/mol between the two transition states of C–S bond dissociation and S–S bond formation [72,76]. This process faced additional challenges posed by two new isotope exchange experiments. These new experiments proved the feasibility of the reverse oxidation of methane [79] and showed a small barrier difference between the two chemical steps [80]. With dispersion and more accurate entropy effects included, a new DFT investigation with methylphenols as tyrosine models gave more reasonable barriers of 15.1 and 21.1 kcal/mol for the forward and reverse reactions, respectively, and a smaller barrier difference of 3.6 kcal/mol for the two transition states [74], which demonstrated that mechanism II is still acceptable and in agreement with mechanistic experiments at that time. Interestingly, an ingenious spectroscopic experiment using a shorter CoB–SH with one methylene less in order to elongate the distance of CoB–SH to the reacting core, lowering the MCR reaction rate, leads to a direct observation of the Ni(II)–SCoM intermediate [81]. This provided the clearest experimental evidence for mechanism II and made mechanism II a consensus MCR mechanism [82].

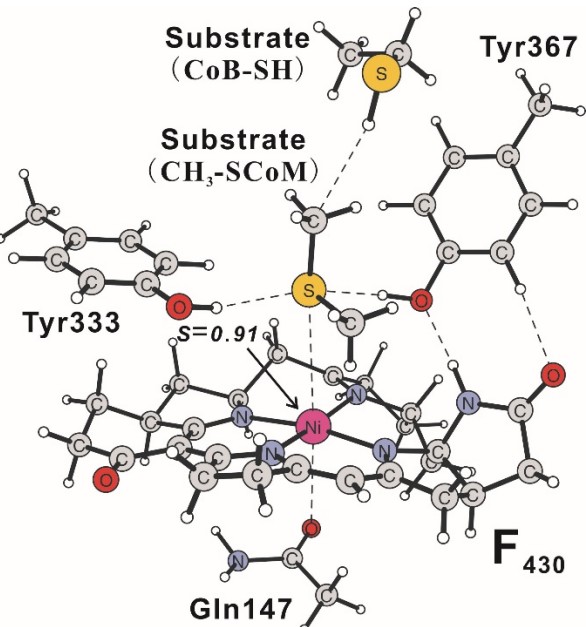

**Figure 6.** Model used in the calculations of the methyl-coenzyme M reductase (MCR). Adapted from [74], copyright © WILEY-VCH Verlag GmbH & Co. KGaA, Weinheim.

An oxidative addition mechanism (i.e., Mechanism III in Figure 5c) proposed by DFT calculations [83] and experiments [80,84] involves a side-on C–S (for $CH_3$–SCoM) [84] or C–H (for the reverse methane oxidation) [80] coordination to the Ni ion, forming a ternary intermediate with both methyl and CoM–S$^-$/hydride coordinating to the metal. However, from a DFT study based on the same cluster model that was used in the previous calculations of mechanism II, the formation of a ternary intermediate was predicted to have a very large endothermicity of 44.5 kcal/mol starting from $CH_3$–SCoM, and also a very high barrier starting from methane [75]. This can safely rule out mechanism

III. Another proposal (mechanism IV in Figure 5d) suggested a protonation at the sulfur position of $CH_3$–SCoM, which should facilitate the C–S bond activation and the subsequent methyl–Ni(III)$F_{430}$ formation. Nevertheless, the DFT calculations showed that the substrate sulfur has a much smaller proton affinity than the $F_{430}$ cofactor, in particular the carbonyl oxygen in $F_{430}$, which means that a proton entering the MCR active site should prefer binding at $F_{430}$ instead of $CH_3$–SCoM, and that an additional energy of ~23 kcal/mol is required to move the proton from $F_{430}$ to the substrate sulfur before the C–S bond activation [75]. This makes mechanism IV infeasible.

In summary, mechanism II (Figure 5b) is finally the most acceptable one for the MCR enzyme after two decades of computational and experimental studies. It fits reasonably well with several structural characteristics of the MCR active site. For example, a deeply-buried active site and the strict arrangement of $F_{430}$, $CH_3$–SCoM, and CoB–SH from bottom to top [12,85], ensuring the quick quenching of the methyl radical. In fact, the absence of CoB–SH cannot initiate the C–S bond cleavage and a CoB–SH analogue with one methylene less makes the MCR reaction much slower [86], which may be explained by the fast rebound between Ni(II)–SCoM and the unstable methyl radical. Even if the methyl radical escapes from CoB–SH, it cannot easily cause irreparable damage to the MCR active site, since the latter is hydrophobic and four active-site residues (His257, Arg271, Gln400, and Cys452) have been uncommonly methylated [12,85]. With this, MCR presents a perfect example of a mechanistic proposal by quantum chemical modeling finally leading to the convergence between theory and experiment. The understanding of the MCR mechanism could lead to new ways of producing methane/alkanes as fuels [87], of utilizing methane/alkanes as chemical feedstocks [82,87], and of breaking the C–S bond [88].

## 4. Nickel CO Dehydrogenase and Acetyl CoA Synthase

Carbon dioxide fixation is an extremely important process in nature, with implications on present day discussions on the greenhouse effect. The dominant enzymes under anaerobic conditions are Ni–CODH and Acetyl CoA synthase, which both have an active site containing Ni. Structures of the enzymes were first determined in 2002–2003 [14,89,90]. In the latest structure [13], obtained at a high resolution of 1.1 Å, an intermediate with bound $CO_2$ was found. The Ni–CODH and Acetyl CoA synthase enzymes are contained in one complex, and are connected by a 140 Å long channel in which CO is transported [91,92]. In Ni–CODH, CO can be reversibly transformed to $CO_2$. The reduction potential used for oxidation of CO is −0.3 V, while for reduction of $CO_2$ it has to be decreased to −0.6 V. Experimentally, the rate for reduction is 10 s$^{-1}$ at pH 7, and at pH 8 the oxidation rate is as high as $3.9 \times 10^4$.

The active site of Ni–CODH has an unusual $NiFe_3S_4$ complex, termed the C-cluster, connected to another iron termed $Fe_u$. The experimental X-ray structure with a bound $CO_2$ [13], termed $C_{red1}$, has carbon bound to nickel and one of the oxygens bound to $Fe_u$, as shown in Figure 7. There are two important hydrogen bonds to $CO_2$, with one being from the positively charged Lys563, and the other one from a His93–Asp219 couple, which is most probably protonated. The spin state is a doublet [93] indicating an oxidation state of Ni(II)Fe$_3$(III)Fe(II) for the NiFe$_4$ cluster. Theoretical studies have supported this assignment [94–96]. After adding two electrons, one C–O bond is cleaved and the state termed $C_{red2}$ is reached, which could be either Ni(I)Fe$_4$(II) or Ni(0)Fe$_3$(III)Fe(II). In the X-ray structure of $C_{red2}$, the CO, resulting from the C–O cleavage of $CO_2$, has disappeared from the cluster leaving an empty site on nickel [13].

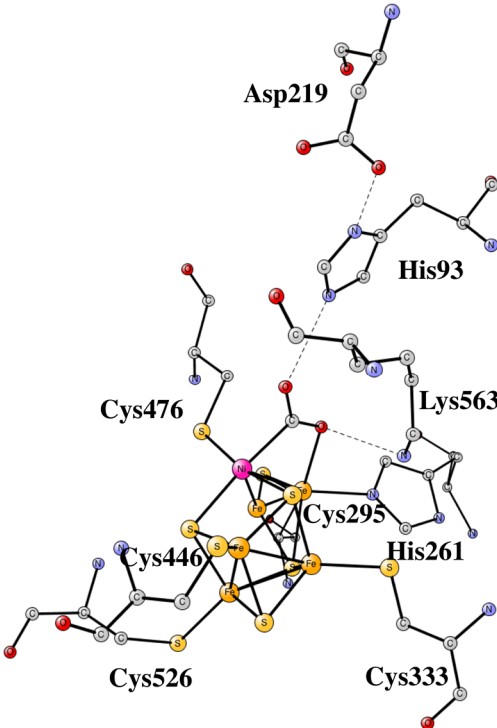

**Figure 7.** The X-ray structure of the active site of Ni–CODH with bound $CO_2$ [13].

The first computational study of the mechanism of Ni–CODH was made by Amara et al. [96]. To explain the X-ray structures, they suggested an unusual mechanism, where in $C_{red2}$ a hydride occupies the empty site on nickel. The hydride would then have a role as an electron reservoir in $C_{red2}$. This would mean that nickel always stays in the Ni(II) state. A bound hydride would not be seen in the X-ray structure and might explain the observation of small amounts of $H_2$ in the experiments. In the $CO_2$ reduction direction from $C_{red2}$, $CO_2$ was suggested to insert into the Ni–hydride bond. The protein matrix should prevent formate formation. Mössbauer parameter calculations, and an orbital and a charge distribution analysis, were also performed to support their mechanism.

Quite recently, the energetics for the full catalytic cycle of Ni–CODH were computed for the first time (Figure 8) [97]. Energy diagrams for both the reduction of $CO_2$, using a redox potential of −0.6 V, and oxidation of CO, using a redox potential of −0.3 V, were calculated. In line with experiments, the oxidation is faster with a barrier difference of 4.8 kcal/mol, compared to experiments of 6.2 kcal/mol [92]. The rate-limiting step in both directions is the cleavage (formation) of the C–O bond of $CO_2$. The strong hydrogen bond from Lys563 and the proton transfer from the His93–Asp219 couple are very important. To obtain accurate energetics, it was found that one of the sulfides in the NiFeS cluster should be protonated. A water molecule in the active site was also found to be reasonably important.

An important difference between the computed and experimental structures found was that CO was bound to nickel in the reduced structure, in contrast to the experimental structure. To explain this result, it was suggested that the X-ray structure was over-reduced. The calculations for the over-reduced structure showed that CO is initially exchanged with a hydride. The hydride and the water molecule then form a hydrogen molecule, suggested to explain the presence of small amounts of $H_2$.

**Figure 8.** Mechanism of nickel CO dehydrogenase suggested on the basis of density functional calculations. Adapted from [97], Copyright © 2019, American Chemical Society.

As mentioned above, acetyl CoA synthase (ACS) is located in the same complex as Ni–CODH, at the other end of the long channel through which CO is transported. The active catalyst in ACS, termed the A-cluster, is a nickel dimer connected by two sulfide bridges to an $Fe_4S_4$ cluster (Figure 9). The nickel closest to the FeS cluster is termed $Ni_p$ and the other one $Ni_u$. $Ni_p$ is, apart from the sulfide bridges to the FeS cluster, connected by two bridging cysteines to $Ni_u$. $Ni_u$ is furthermore ligated to two backbone nitrogens of two connected cysteines. $Ni_p$ obtains a square planar coordination with a Ni(II) oxidation state before the substrates enter. $Ni_p$ is five-coordinated and has been suggested to be the active metal in the dimer. There are three substrates, CO, $CH_3$, and coenzyme A (CoA). $CH_3$ is donated to the cluster from a cobalamine. The order of the entering substrates is debated. There are reasons to believe that the order might matter, since the protein has at least two different conformations, of which only one—the open conformation—allows CO to be transported all the way to ACS. In the closed conformation the transport of CO is blocked.

There has been one theoretical study of the mechanism of ACS [98]. A cluster QM approach was used. A problem in the modeling is that no X-ray structures with bound substrates are available. Possible intermediates in the catalytic cycle were studied, including their oxidation states. Ni(0) had earlier been suggested as a possible state of $Ni_p$, but the study shows that this is unlikely. The zero-valent state is stabilized only if the FeS cluster is reduced to a state with S = 1/2 ($[Fe_4S_4]^{1+}$). As CO binds, $Ni_p(0)$ is still favored. If the FeS cluster is oxidized, $Ni_p(I)$ is instead favored. It is suggested that CO and $CH_3$ both bind to $Ni_p$ in a mononuclear mechanism on $Ni_p$. There are major remaining issues in the mechanism, mostly because of the limited structural information. In particular, the transfer of the

methyl from cobalamine to the nickel dimer would be very interesting to study in the future, but also many questions remain regarding the final stages of CoA synthase.

**Figure 9.** Mechanism of Acetyl CoA suggested on the basis of quantum mechanics (QM) cluster calculations. Adapted from [98], Copyright © 2005, American Chemical Society.

A theoretical study of the Ni-dimer in ACS with emphasis on pK$a$ values and redox potentials has also been performed [99]. Based on these computed values, the initial part of a mechanism for ACS was suggested. The low reduction potential computed indicates that the reduced form is protonated. Before or during methylation of the A-cluster, the proton could be detached by a base. A proton coupled electron transfer (PCET) mechanism should be operational in the methylation step. Furthermore, a short early note on the ACS mechanism by another group has been published [100]. Their calculations suggest that $Ni_p$ is better described by the oxidation state Ni(I) than Ni(0).

## 5. Acireductone Dioxygenase

Acireductone dioxygenases (ARDs) are a family of metalloenzymes that are capable of using different transition metal ions ($Ni^{2+}$ and $Fe^{2+}$) to catalyze the dioxygenation of acireductone [101,102]. $Fe^{2+}$-ARD and $Ni^{2+}$-ARD share exactly the same protein sequence. However, $Fe^{2+}$-ARD catalyzes the formation of methylthioketobutyrate and formate [101], whereas $Ni^{2+}$-ARD oxidizes acireductone into three products, namely formate, 3-methylthiopropionic acid, along with carbon monoxide [102]. It has been previously suggested that the substrate binding mode for $Fe^{2+}$-ARD is a five-membered ring,

while there is a six-membered ring for $Ni^{2+}$-ARD (Scheme 1) [103], and this should be the main reason for the formation of different products.

**Scheme 1.** The previously suggested mechanisms for ARDs. Adapted from [103], Copyright © 2001, American Chemical Society.

Sparta et al. performed discrete molecular dynamics simulations (QM/DMD) to investigate the hypothesis of the different substrate binding modes in $Fe^{2+}$-ARD and $Ni^{2+}$-ARD [104]. The QM/DMD simulations suggested that both $Fe^{2+}$-ARD and $Ni^{2+}$-ARD exhibit a six-membered ring coordination, with the O1 and O3 atoms binding to the central metal ion (Scheme 2, React). A second-shell residue Arg154 forms a hydrogen bond to the O2 atom of the substrate, which stabilizes this coordination mode. One snapshot was then selected for DFT calculations in order to understand the mechanisms of $Fe^{2+}$-ARD and $Ni^{2+}$-ARD. The energy barrier for the first dioxygen attack was not considered. The starting complex is a dioxygen adduct, with a peroxy bridging C1 and C3 of the substrate (reaction, Scheme 2). For $Ni^{2+}$-ARD, the reaction takes place in the triplet state. After a concerted cleavage of the O–O, C1–C2, and C2–C3 bonds, carbon monoxide and two carboxylates are generated (production, Scheme 2). The energy barrier is only 3.7 kcal/mol.

**Scheme 2.** Reaction mechanism for Ni-acireductone dioxygenase supported by the results of a DFT study. Adapted from [104], Copyright © 2013 Elsevier Ltd.

Recently, based on the structure of human acireductone dioxygenase, Borowski et al. have performed QM/MM calculations to revisit the reaction mechanisms of ARDs [105]. For the $Ni^{2+}$-ARD, the calculations suggested that the mechanism involves four major steps (Scheme 3). A proton is first transferred from the substrate to His88 when the substrate enters into the active site, and His88 dissociates from the nickel ion. Dioxygen binds to the metal coupled with one electron transfer from the substrate to the dioxygen moiety to generate a $Ni^{2+}$-superoxide intermediate (complex **1**), which is quite similar to quercetin 2,3-dioxygenases [106,107]. The ground state of complex **1** is a quintet state (high-spin nickel(II) ions, one electron on the dioxygen, one electron on the substrate), while the following reaction is in the triplet state (two unpaired electrons on the nickel (II) ion). Hence, the energy of the triplet complex **1** was taken as the reference. After the distal oxygen atom attacks the C2

atom, a peroxo intermediate (complex **2**) is formed. This step was found to be barrier-less. The next step is the migration of the C2-bound oxygen atom to the C3 atom (complex **3**), which is ready for the following formation of the O–O bridge between C1 and C3 (complex **4**). The final concerted cleavages of the O–O, C1–C2, and C2–C3 bonds were found to be the rate-limiting steps (TS45) for the substrate oxidation, with a barrier of only 6.8 kcal/mol. It is more likely that the substrate binding or product release is the rate-limiting step for the whole reaction.

**Scheme 3.** Reaction mechanisms for Ni-ARD supported by the results of a quantum mechanics/molecular mechanics (QM/MM) study. Adapted from [105], copyright © WILEY-VCH Verlag GmbH & Co. KGaA, Weinheim.

Alternative pathways have also been considered from complex **2**. The C2-bound oxygen atom may attack the C1 atom to form complex **6**. After the formation of a dioxethane intermediate (complex **7**) and the cleavage of the O–O and C1–C2 bonds (TS78), methylthioketobutyrate and formate are generated (complex **8**). However, the barrier of TS78 was found to be 11.2 kcal/mol, higher than that of TS45. Other pathways that pass through a Baeyer-Villiger type rearrangement (**2** → **6** → **10** → **11**) and the homolytic O–O cleavage (**2** → **6** → **9**) were also ruled out due to their higher energy barriers compared with the most favorable mechanism.

## 6. Quercetin 2,4-Dioxygenase

Quercetin 2,4-dioxygenase from *Streptomyces* sp. strain FLA is a nickel-dependent enzyme catalyzing the oxidative ring-cleavage of quercetin to produce CO and 2-protocatechuoylphloroglucinol carboxylic acid using $O_2$ as the oxidant [16,108]. The mechanism for the dioxygen activation and substrate oxidation was first investigated by Liu and co-workers [109]. From the 19 ns molecular dynamics simulations, one snapshot was chosen for the following QM/MM calculations. The first-shell ligand Glu76 set to be deprotonated as a proton is transferred from the neutral quercetin substrate. For the reactant complex with the $O_2$ binding to the Ni ion in an end-on fashion, their calculations showed that the quintet is the ground state, the triplet lies 2.3 kcal/mol higher in energy, while the singlet was found to be much higher in energy. It is likely that the closed-shell singlet was obtained, while the open-shell singlet was not considered. In the triplet state, one electron is transferred from the substrate to the $O_2$ moiety, leading to the formation of a $Ni^{2+}$-superoxide-substrate radical species. Importantly, the reaction takes place in the triplet state involving four major steps (Scheme 4). First, the terminal

oxygen of the superoxide attacks C2 of the substrate to form the first C–O bond, which has a barrier of only 5.0 kcal/mol. This is followed by a conformational change by a rotation around the newly-formed C2–O bond, and this step turned out to be rate-limiting, with a barrier of 19.9 kcal/mol. Subsequently, the terminal oxygen of the peroxide attacks C4 of the substrate, resulting in a peroxide-bridging intermediate. Finally, simultaneous cleavage of the O–O bond and two C–C bonds produces a CO molecule and the 2-protocatechuoylphloroglucinol carboxylic acid. When the central nickel ion is replaced by an iron ion, the reaction proceeds via a similar pathway in the quintet state. However, the last step becomes rate-limiting, associated with a very high barrier of 29.9 kcal/mol.

**Scheme 4.** Reaction mechanism for Ni-QueD suggested by Liu et al. Adapted from [109], with permission from The Royal Society of Chemistry.

Liao and co-workers independently investigated the reaction mechanism of this enzyme almost at the same time [107]. In their QM/MM calculations, the model started directly from the X-ray conformation. Importantly, the first-shell ligand Glu74 was considered to be both neutral and deprotonated. The aim of that study was to elucidate the mechanism and also to rationalize the chemoselectivity, in which only 2,4-dioxygenolytic cleavage takes place, but not 2,3-dioxygenolytic cleavage. They found a similar mechanism as Liu et al. [109] when Glu74 is protonated. However, the last step was found to be rate-limiting with a barrier of 24.8 kcal/mol. Unexpectedly, the 2,3-dioxygenolytic pathway was found to be more favorable, with a barrier of 21.8 kcal/mol. Therefore, the model with a protonated Glu74 residue could not reproduce the observed chemoselectivity. Instead, a model with a deprotonated Glu74 was demonstrated to reproduce the chemoselectivity (Scheme 5). In this case, the total barrier for the 2,4-dioxygenolytic pathway decreases to 17.4 kcal/mol, while it increases to 30.6 kcal/mol for the 2,3-dioxygenolytic pathway. The calculated barrier of 17.4 kcal/mol is in good agreement with the experimental kinetic data [108], which gave a barrier of about 15 kcal/mol.

**Scheme 5.** Reaction mechanism and chemoselectivity for Ni-QueD suggested by Liao et al. Adapted from [107], with permission from the PCCP Owner Societies.

## 7. Urease

Urease is a nickel-containing enzyme that catalyzes the hydrolysis of urea to yield ammonia and carbon dioxide in the last step of nitrogen mineralization [110]. The uncatalyzed reaction has been suggested by Merz and Estiu [111] to proceed via an elimination pathway. For the enzyme catalyzed reaction, there have been three different mechanistic proposals. First, Karplus et al. suggested a mechanism in which a Nickel-coordinated terminal hydroxide performs the nucleophilic attack, while an adjacent protonated His320 residue delivers a proton to the leaving amino group [110,112]. Second, Benini et al. proposed a mechanism involving a nucleophilic attack of a bridging hydroxide on the carbonyl carbon of urea, followed by the protonation of the leaving amino group by this hydroxide [113]. Third, Lippard et al. suggested an elimination mechanism, similarly to the uncatalyzed reaction [114]. However, this elimination pathway could not explain the experimental observation of carbamate as the first product found by Zerner et al. [115]. A number of computational studies have, therefore, been conducted to elucidate the reaction mechanism of this enzyme [116–123].

The two different hydrolytic mechanisms were investigated by Merz et al. using DFT calculations (Schemes 6 and 7) [120]. For the bridging hydroxide attack mechanism, the calculations suggested that the reaction starts from a $Ni^{II}_2$ complex with a bridging hydroxide (Scheme 6). In the reactant complex S3, the urea substrate binds in a bi-dentate fashion, where the carbonyl oxygen coordinates to Ni1, while one of the amino groups coordinate to Ni2. Both $Ni^{II}$ ions are in the triplet state, and the complex is, thus, a quintet for the ferromagnetically-coupled state. The bridging hydroxide performs a nucleophilic attack on the carbonyl carbon of the urea substrate via TS1A, leading to the formation of a tetrahedral intermediate (I1A) with the oxyanion stabilized by one of the two nickel ions. This pathway is consistent with the proposal by Ciurli et al., on the basis of docking studies [120]. Then, a proton is delivered from the bridging hydroxide to the amino group of urea (TS2A), facilitated by Asp360. This leads to the production of an ammonia molecule and a carbamate. The second step was found

to be rate-limiting, with a barrier of 19.7 kcal/mol. Protonation of the leaving ammonia group by a protonated His320 residue has also been taken into account, and this step is close to isothermic. It should be noted that the role of this histidine residue has very recently been confirmed by modulating its protonation state and conformation of the mobile flap [124]. After the release of the ammonia molecule, a water molecule binds to Ni2, and this water molecule then performs a nucleophilic attack on the carbon atom of the carbamate anion (TS4A), concertedly with a proton transfer from the water molecule to the nearby Asp360 residue. Finally, the protonated Asp360 residue delivers a proton to the bridging oxyanion (TS6A), coupled with the C–O bond cleavage. The di-nickel complex (I7A) with a bridging hydroxide is regenerated. It should be pointed out that this kind of mechanism agrees very well with the recent crystal structure of the complex between urea and fluoride-inhibited urease, in which an unreactive fluoride replaces the bridging hydroxide [125].

**Scheme 6.** The bridging hydroxide attack mechanism suggested by Merz et al. on the basis of DFT calculations. Adapted from [120], Copyright © 2003, American Chemical Society. Energies are given in kcal/mol relative to S3.

**Scheme 7.** The nickel bound water nucleophilic attack mechanism considered by Merz et al. on the basis of DFT calculations [120], Copyright © 2003, American Chemical Society. Energies are given in kcal/mol relative to S3 in Scheme 6.

Nucleophilic attack by the nickel bound water molecule was also taken into consideration by Merz et al. [120], and the starting complex was a monodentate urea-bound adduct with the carbonyl oxygen coordinated to Ni1 (S1B, Scheme 7). The calculations suggested that this mechanism involves only two steps (Scheme 7): the Ni2-bound water molecule performs a nucleophilic attack on the urea carbonyl carbon, which is coupled with a proton transfer from the water molecule to the leaving ammonia, assisted by the bridging hydroxide (TS1B). The next step is the C–N bond cleavage with the release of an ammonia molecule. The second step is rate-limiting, with a barrier of 24.6 kcal/mol, which is higher than the one of the bridging attack by a hydroxide.

They have also performed DFT calculations to analyze the elimination mechanism of urease (Scheme 8) [122]. They started from the enzyme without the urea substrate, and the substrate binding was calculated to be endothermic by 5.0 kcal/mol. Then, deprotonation of the amino group of urea by the bridging hydroxide should take place (TS3uw) and a new bridging water molecule is generated (I3uw). In the next step, a proton is transferred from the bridging water to the other amino group of urea (TS4uw), with a barrier of 30.5 kcal/mol. Subsequently, a water molecule enters into the active site and forms hydrogen bonds with the two Ni-bound water molecules, which facilitates the release of $NH_4^+$ and $OCN^-$. This mechanism has a substantially higher barrier than those for the two other mechanisms.

**Scheme 8.** The elimination mechanism of urea considered by Merz et al. on the basis of DFT calculations. Adapted from [122], Copyright © 2007, American Chemical Society.

## 8. Lactate Racemase

Lactate racemase (LarA) has been identified to be the enzyme that catalyzes the final step in the racemization between D- and L-lactic acids [126], using a previously unknown Ni cofactor (Ni-PTTMN), pyridinium-3-thioamide-5-thiocarboxylic acid mononucleotide Ni pincer complex [19,127] (Figure 10). Ni-PTTMN is formed by the coordination between a Ni(II) ion and a prosthetic group derived from nicotinic acid (Figure 10). The latter has a thioamide (modified from the nicotinic moiety) covalently bound to Lys184 and a thiocarboxylate, and acts as a tridentate pincer ligand with the two sulfurs and one pyridine carbon ligating to the nickel, where a Ni–C bond is formed. The His200 works as the fourth nickel ligand. The Ni-PTTMN cofactor in LarA serves as a rare example of a pincer complex found in enzymes and its Ni–C bond is the first case of a metal-carbon bond originating from nicotinic

acid in biology [128]. Considering the common and widespread NAD$^+$-type cofactors (nicotinamide adenine dinucleotide and its phosphorylated form, NADP$^+$) derived from nicotinic acid, it is surprising that such a complicated Ni-PTTMN cofactor has been biologically evolved with extra consumption of resources. A question is whether it has evolutional advantages compared with NAD$^+$ [128].

**Figure 10.** A modified proton-coupled hydride transfer mechanism for lactate racemase (LarA) proposed by Chen et al. Adapted from [129], copyright © WILEY-VCH Verlag GmbH & Co. KGaA, Weinheim. Tyr294 and Lys298 play crucial roles in orienting substrates and stabilizing the negative charge developing at the substrate hydroxyl oxygen in the transition states, thus lowering reaction barriers significantly.

A proton-coupled hydride transfer (PCHT) mechanism using a Ni(II) ion (Figure 10) has been verified by two DFT cluster modeling investigations by Chung et al. [130] and Chen et al. [129]. In this mechanism, a hydride is transferred to the Ni-PTTMN pyridine carbon from the substrate $\alpha$-carbon, coupled with a proton transfer from the substrate hydroxyl to a histidine, followed by a transfer back to the $\alpha$-carbon of the resultant pyruvate from the opposite side. This is achieved via a rotation of the pyruvate acetyl moiety. Chung et al. used an active-site model of 139 atoms to give a barrier of ~18 kcal/mol for the hydride transfer at the B3LYP-D3 level [130]. Based on small models (mainly involving cofactors and a proton-acceptor imidazole) and low-polarity solvation effects with a dielectric constant ($\varepsilon$) of 4, the further comparison of Ni-PTTMN with NAD$^+$-like cofactors showed that the Ni-PTTMN cofactor has a higher racemization barrier than NAD$^+$ [130], which makes it interesting to explore the LarA activity in the full active-site environment. In the subsequent DFT calculations by Chen et al. using a 200-atom model with Tyr294 and full side chains of the residues included, a lower rate-limiting barrier of 12 kcal/mol for the hydride transfer was predicted. The catalytic acceleration effect is achieved via the stabilization of the transition states by Tyr294 and Lys298 [129]. Further calculations were performed with various modified Ni-PTTMN cofactors in the

LarA active site. They showed that the barrier of 12 kcal/mol for the native LarA with a Ni-PTTMN cofactor is much lower than those (at least by 19 kcal/mol) for the LarA enzymes with NAD$^+$-like cofactors [129], indicating an enhanced racemization activity of Ni-PTTMN. To further analyze the acceleration effects of Ni-PTTMN, the hydride affinities of the Ni-PTTMN and NAD$^+$-like cofactors were calculated [129]. It was revealed that compared with NAD$^+$-like cofactors, Ni-PTTMN has a hydride affinity less sensitive to the environmental polarity, thus keeping a stronger hydride-addition reactivity in moderately and highly polar surroundings ($\varepsilon \geq 8$), while a weaker one in a low-polarity environment ($\varepsilon < 8$) [129]. This not only explains why a higher barrier for Ni-PTTMN was obtained in the calculations by Chung et al. [130], but also indicates the evolutionary advantage of Ni-PTTMN using an architecture with a Ni pincer, which fits perfectly in the moderately polar active site of LarA, in which a dielectric constant of around 14–18 was estimated by a barrier analysis [130]. Inspired by the Ni-PTTMN-dependent LarA reaction, a series of Ni-PTTMN-like metal pincer complexes (metal = Ni and Pd) were proposed by Yang et al. using DFT calculations to be potential catalysts for lactate racemization with a barrier as low as ~26 kcal/mol, adopting the same PCHT mechanism as LarA [131], which is a step closer to utilizing the novel Ni-PTTMN cofactor and its related chemistry in man-made catalysis.

Another mechanism starting from a Ni(III) state was proposed in a QM/MM study by Shaik et al. [132]. It includes the electron transfer from the lactate carboxylate to Ni(III), the C–C bond dissociation leading to a $CO_2$ radical anion and an acetaldehyde, the rotation of acetaldehyde, and then a C–C rebound step and the electron transfer back to the carboxylate. In this mechanism, the Ni-PTTMN cofactor works as a reversible electrode to accept and donate back an electron and does not result in covalent bonding to the lactate moiety. However, it is not clear if a Ni(II) Ni-PTTMN cofactor can be oxidized to the Ni(III) state in the LarA active site [133].

## 9. Superoxide Dismutase

Superoxide dismutases (SOD) catalyze the disproportionation of the toxic superoxide radicals into the less toxic hydrogen peroxide and molecular oxygen,

$$2O_2^- \cdot + 2H^+ \rightarrow O_2 + H_2O_2 \tag{2}$$

They have an important protective role, particularly in aerobic metabolism and photosynthesis. There are at least three different types of SOD's, and in one of them there is an active nickel complex.

The metal complex in Ni-SOD has a mononuclear nickel, coordinated to two cysteines and two backbone nitrogens in a square planar configuration (Figure 11). This is very similar to the coordination of $Ni_u$ in acetyl CoA synthase (see above). However, there is one additional, axial, histidine ligand, which stabilizes a Ni(III) oxidation state. The disproportionation reaction occurs in two consecutive half-reactions, each one with one superoxide radical substrate. In the oxidative phase, the first superoxide radical binds and is transformed to dioxygen,

$$Ni(III)\text{-}L + O_2^- \cdot + H^+ \rightarrow Ni(II)\text{-}LH + O_2 \tag{3}$$

In the reductive phase, the second superoxide substrate binds and is transformed to hydrogen peroxide,

$$Ni(II)\text{-}LH + O_2^- \cdot + H^+ \rightarrow Ni(III)\text{-}L + H_2O_2 \tag{4}$$

Shortly after the first high-resolution X-ray structure appeared in 2004 [17], there were two independent quantum chemical studies of the mechanism [134,135]. Essentially two different mechanisms had been suggested based on experiments. The most popular one was based on the findings on the structure formed by X-ray reduction [17,136]. It was found that upon reduction, the axial histidine lost its coordination to nickel. Therefore, it was suggested that the His became protonated as the proton mediator in the mechanism. The alternative is that one of the cysteines

becomes protonated instead. The calculations used essentially minimal cluster models, including only the directly coordinating ligands and the backbone connecting the two cysteines.

The mechanism obtained in the first study [134] is shown in Figure 11. It was suggested that the superoxide substrate enters as an $O_2H\cdot$ radical and binds in the empty axial site, where the cost for the protonation was included in the energetics. Anti-ferromagnetic coupling between nickel and the radical was found to be best. After substrate binding, the proton moves over to Cys2 and $O_2$ leaves. Nickel has now been reduced to Ni(II). The second substrate radical then enters and binds at the same axial position. The proton on Cys2 is now well suited to move over to the $O_2H^-$ ligand to form $H_2O_2$, and the disproportionation is completed. The protonation of His1 was found to be much less favorable. The barrier for the first half-reaction was found to be 9.7 kcal/mol and for the second one 11.5 kcal/mol, in good agreement with expectations based on experiments.

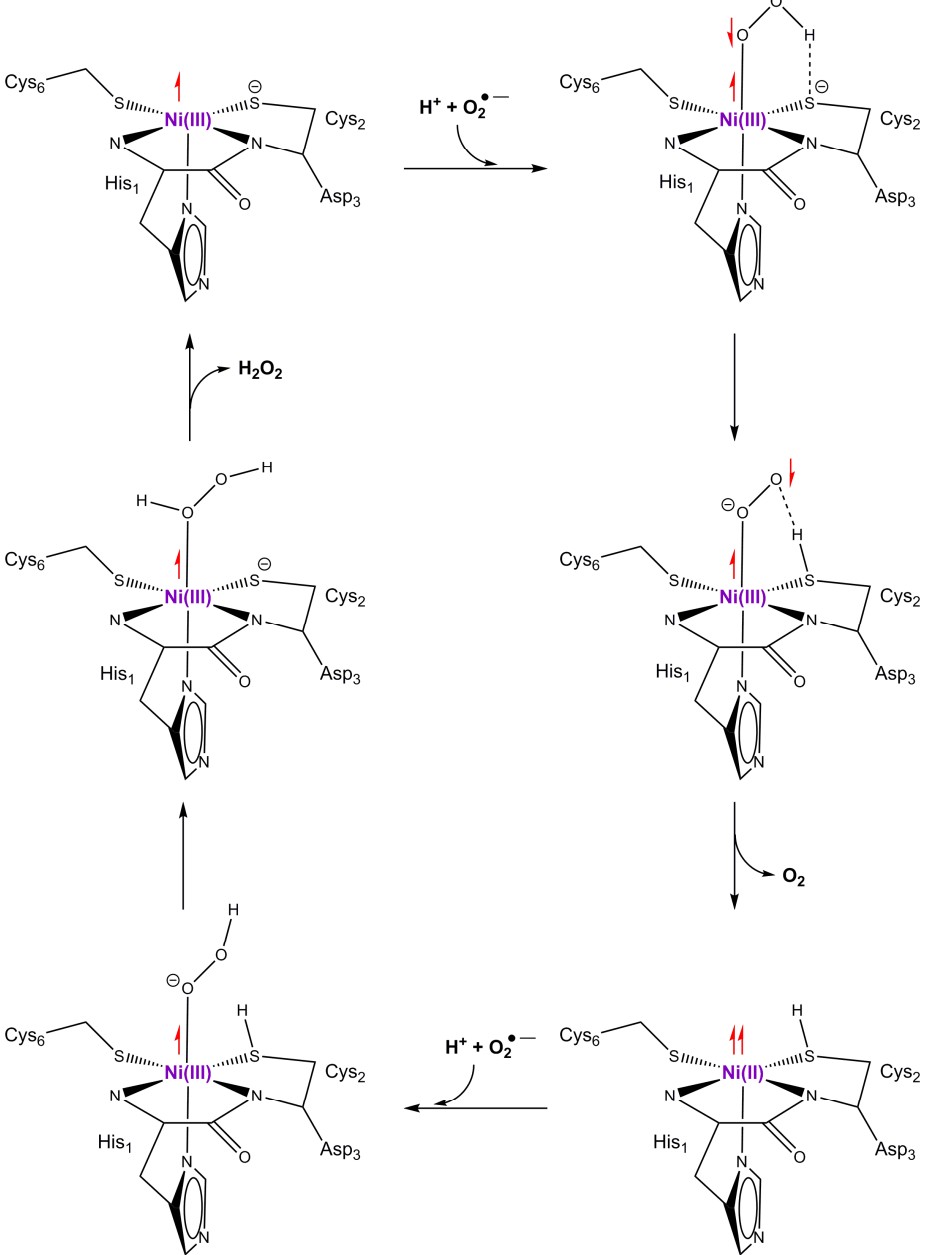

**Figure 11.** Reaction mechanism for superoxide dismutase (Ni-SOD) suggested by Pelmenschikov et al. Adapted from [134], Copyright © 2006, American Chemical Society.

In the second study, a different mechanism was found for the first half-reaction, which was the only one studied [135]. A few second shell residues were included in the model apart from the directly binding ones. It was suggested that Cys6 started out protonated and donated its proton to the first superoxide. $H_2O_2$ can then be formed by a donation of a proton from the medium via His1 and Tyr9. A proton donating role of His1 was, therefore, suggested in line with earlier experimental suggestions, based on the X-ray reduced structure [17,136]. Transition states were not determined due to the large size of the model.

Two additional studies can be mentioned. Several years later in 2011, a study was made [137] that reached similar conclusions as the earlier ones. An energy diagram was calculated showing a rate-limiting step with a barrier for one of them of around 19 kcal/mol. In another study on a mimic of SOD, multireference effects were studied with the use of the complete active space self-consistent field method (CASSCF) [138]. It was found that the near-degeneracy effects were small, which was considered surprising at the time. Multi-reference effects were expected, since a radical became bound to a transition metal.

## 10. Conclusions

In this review, we have presented the progress of theoretical studies on the reaction mechanisms of nickel-dependent enzymes. Both quantum chemical cluster and QM/MM approaches have been successfully used to answer mechanistic questions in these enzymes. Detailed information can be obtained on the structures, energies, electron densities, and spin densities of all relevant intermediates and transition states.

Nickel is redox active and the common oxidation states are $Ni^I$, $Ni^{II}$, and $Ni^{III}$ in Ni enzymes. Due to its flexible coordination geometry, its redox potential can be tuned by different ligand coordinations with a span from +0.89 V to −0.60 V. Therefore, it can be used for challenging redox reactions, such as for superoxide dismutase, selective reduction of $CO_2$, and methane formation. In addition, the nickel ion can also function as a Lewis acid to stabilize anionic intermediates and transition states during the reactions.

**Author Contributions:** Writing—original draft preparation, P.E.M.S., S.-L.C., and R.-Z.L.; writing—review and editing, P.E.M.S., S.-L.C., and R.-Z.L.

**Funding:** This research was funded by the Swedish Research Council, the Knut and Alice Wallenberg Foundation, the National Natural Science Foundation of China, grant numbers 21673019 and 21873031.

**Conflicts of Interest:** The authors declare no conflict of interest.

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
