# Peer review of "Theoretical Studies of Nickel-Dependent Enzymes"

_inorganics, doi:10.3390/inorganics7080095_

Round 1

Reviewer 1 Report

This review article provides an overview of computational studies on reaction mechanisms of nickel dependent enzymes. The Authors succeded in extracting from the original papers the key findings about the reaction mechanisms and presented them in a clear and systematic way. A synthesis of the reaction patters and catalytic strategies used by these enzymes has also been provided.  I believe this review will be an important reference point for anyone interested in nickel chemistry and biochemistry. 

Minor points / typos to be corrected in the revision:

- page 6, line 149: a label 'Ni-L' is introduced but not explained

- page 12, line 317: 'Acetyl CoA (ACS)', this probably should read 'Acetyl CoA Synthase (ACS)' 

- page 12, line 333: the statement 'if the FeS cluster is reduced with s=1/2' is a bit vague; what does the 's=1/2' refer to ?

- page 17, Scheme 5: the reaction arrows between Prod_B and React and Prod_A and React are misleading as they do not report product release and substrates binding 

-page 17, line 441: 'pathway For the enzyme' - full stop missing ?

Author Response

This review article provides an overview of computational studies on reaction mechanisms of nickel dependent enzymes. The Authors succeded in extracting from the original papers the key findings about the reaction mechanisms and presented them in a clear and systematic way. A synthesis of the reaction patters and catalytic strategies used by these enzymes has also been provided.  I believe this review will be an important reference point for anyone interested in nickel chemistry and biochemistry. 

Minor points / typos to be corrected in the revision:

- page 6, line 149: a label 'Ni-L' is introduced but not explained.

Reply:  “Ni-L” in Ryde et al’s paper corresponds to “Nia-R*” in our Figure 3. This is now revised.

- page 12, line 317: 'Acetyl CoA (ACS)', this probably should read 'Acetyl CoA Synthase (ACS)' 

Reply:  This is now corrected.

- page 12, line 333: the statement 'if the FeS cluster is reduced with s=1/2' is a bit vague; what does the 's=1/2' refer to ?

Reply:  This is now corrected into “The zero-valent state is stabilized only if the FeS cluster is reduced to a state with S=1/2 ([Fe4S4]1+)”.

- page 17, Scheme 5: the reaction arrows between Prod_B and React and Prod_A and React are misleading as they do not report product release and substrates binding 

Reply: This is now corrected, and the arrows between Prod and React are removed.

-page 17, line 441: 'pathway For the enzyme' - full stop missing ?

Reply: A full stop is added.

Reviewer 2 Report

The review is a nice, comprehensive and yet concise contribution to the scientific community that focuses on the role of nickel in biological systems, and I am sure it will represent a key reference paper. I have only a few suggestions for the authors, and both concern the urease chapter.

1) The proposal by Benini et al. was not the "first" as it was published in 1999, while the proposal by Hausinger was initially put forth in 1997, and reiterated in 2000.

2) The elimination mechanism by Lippard was based on a small molecule complex, and its reliability is very limited due to the much simpler system on which it was based, as compared to the enzyme. Moreover, it should be stated that formation of products of elimination was excluded by Zerner et al. years before (Biochemistry 1969, 8, 1991 – 2000), and therefore the elimination mechanism should be excluded, unless it is stated that it has nothing to do with urease and is limited only to small molecules. In this context, the reference for the Lippard and Barrios paper is wrong (it is a "5", not formatted as reference, and the number is also wrong).

3) Consequently, while it might be correct to report that Merz et al. have nevertheless considered this possibility in their theoretical studies, it should also be stated that those calculations are meaningless in the framework of the enzyme itself.

4) As for the two hydrolytic mechanisms proposed by Benini et al. and by Hausinger et al., the authors should comment on two very recent papers published in Angew. Chem. 2019, 131, 7493–7497 and in Chem. Eur. J. in press (DOI 10.1002/chem.201902320). In the first paper, the crystal structure of the complex between urea and fluoride-inhibited urease, in which an un-reactive fluoride replaces the bridging hydroxide, is described, providing full support of the proposal by Benini et al. and excluding definitively the proposal by Haousinger, while in the second paper the important role of a mobile flap gating the active site cavity is discussed in the framework of the dependence of its conformation on solution pH, an effect hardly reproducible by QM or QM/MM calculations.

5) In the list of theoretical studies on the mechanism of urease, the following reference is missing, and should also be discussed:J Biol Inorg Chem 2001, 6 (3), 300–314.

Author Response

The review is a nice, comprehensive and yet concise contribution to the scientific community that focuses on the role of nickel in biological systems, and I am sure it will represent a key reference paper. I have only a few suggestions for the authors, and both concern the urease chapter.

1) The proposal by Benini et al. was not the "first" as it was published in 1999, while the proposal by Hausinger was initially put forth in 1997, and reiterated in 2000.

Reply:  This is now revised in page 16. “First, Karplus et al. suggested a mechanism, in which a Nickel-coordinated terminal hydroxide performs the nucleophilic attack, while an adjacent protonated His320 residue delivers a proton to the leaving amino group [115,117]. Second, Benini et al. proposed a mechanism involving a nucleophilic attack of a bridging hydroxide on the carbonyl carbon of urea, followed by the protonation of the leaving amino group by this hydroxide [118].

2) The elimination mechanism by Lippard was based on a small molecule complex, and its reliability is very limited due to the much simpler system on which it was based, as compared to the enzyme. Moreover, it should be stated that formation of products of elimination was excluded by Zerner et al. years before (Biochemistry 1969, 8, 1991 – 2000), and therefore the elimination mechanism should be excluded, unless it is stated that it has nothing to do with urease and is limited only to small molecules. In this context, the reference for the Lippard and Barrios paper is wrong (it is a "5", not formatted as reference, and the number is also wrong).

Reply: It is now corrected.  A comment on this mechanism based on Zerner’s work is now added in page 16. “However, this elimination pathway could not explain the experimental observation of carbamate as the first product found by Zerner et. al.[120].

3) Consequently, while it might be correct to report that Merz et al. have nevertheless considered this possibility in their theoretical studies, it should also be stated that those calculations are meaningless in the framework of the enzyme itself.

Reply: We found that Merz et al. investigated this mechanism, and this is discussed in page 18 (reference 126).

4) As for the two hydrolytic mechanisms proposed by Benini et al. and by Hausinger et al., the authors should comment on two very recent papers published in Angew. Chem. 2019, 131, 7493–7497 and in Chem. Eur. J. in press (DOI 10.1002/chem.201902320). In the first paper, the crystal structure of the complex between urea and fluoride-inhibited urease, in which an un-reactive fluoride replaces the bridging hydroxide, is described, providing full support of the proposal by Benini et al. and excluding definitively the proposal by Haousinger, while in the second paper the important role of a mobile flap gating the active site cavity is discussed in the framework of the dependence of its conformation on solution pH, an effect hardly reproducible by QM or QM/MM calculations.

Reply: These are now discussed in page 17. “It should be noted that the role of this histidine residue has very recently been confirmed by modulating its protonation state and conformation of the mobile flap [129].” “It should be pointed out that this kind of mechanism agrees very well with the recent crystal structure of the complex between urea and fluoride-inhibited urease, in which an unreactive fluoride replaces the bridging hydroxide [130].

5) In the list of theoretical studies on the mechanism of urease, the following reference is missing, and should also be discussed: J Biol Inorg Chem 2001, 6 (3), 300–314.

Reply: This is now discussed briefly in page 16. “This pathway is consistent with the proposal by Ciurli et. al, on the basis of docking studies[124].